# Computer-Based Simulators in Pharmacy Practice Education: A Systematic Narrative Review

**DOI:** 10.3390/pharmacy11010008

**Published:** 2023-01-02

**Authors:** Ahmed M. Gharib, Ivan K. Bindoff, Gregory M. Peterson, Mohammed S. Salahudeen

**Affiliations:** School of Pharmacy and Pharmacology, University of Tasmania, Hobart, TAS 7005, Australia

**Keywords:** computer-based simulation, pharmacy practice education, pharmacy education, online simulation, virtual simulation, virtual patient, and case-based simulation

## Abstract

Computer-based simulations may represent an innovative, flexible, and cost-efficient training approach that has been underutilised in pharmacy practice education. This may need to change, with increasing pressure on clinical placement availability, COVID-19 restrictions, and economic pressures to improve teaching efficiency. This systematic narrative review summarises various computer-based simulations described in the pharmacy practice education literature, identifies the currently available products, and highlights key characteristics. Five major databases were searched (Medline, CINAHL, ERIC, Education Source and Embase). Authors also manually reviewed the publication section of major pharmacy simulator websites and performed a citation analysis. We identified 49 studies describing 29 unique simulators, which met the inclusion criteria. Only eight of these simulators were found to be currently available. The characteristics of these eight simulators were examined through the lens of eight main criteria (feedback type, grading, user play mode, cost, operational requirement, community/hospital setting, scenario sharing option, and interaction elements). Although a number of systems have been developed and trialled, relatively few are available on the market, and each comes with benefits and drawbacks. Educators are encouraged to consider their own institutional, professional and curriculum needs, and determine which product best aligns with their teaching goals.

## 1. Introduction

Computer-based simulation (CBS) in healthcare education can be defined as a virtual simulation of clinical scenarios that allows users to replicate the role of a health professional in a given situation [1,2]. Studies have demonstrated that CBS can provide a nexus between theory and practice [2,3,4,5,6,7]. CBS encourages students to develop their clinical decision-making using evidence-based practice in an active learning environment and test their knowledge, in contrast to traditional memorisation-based education [8,9]. It can be used as both a learning and practical assessment tool [10]. When compared to traditional instructional design, CBS stands out as a more scalable, accessible and cost-efficient educational tool, that offers a standardised experience for pharmacy students [11,12,13].

Many pharmacy education providers have been inspired to implement, study, and improve CBS training tools in their curricula [14,15]. CBS has complemented and even changed the training methods used in pharmacy schools globally, with engagement and learning outcomes comparable with [16,17] or exceeding conventional methods [18,19,20]. 

Studies have explored the general usefulness of CBS as a teaching method (measuring the satisfaction of targeted users) or validated the use of a specifically developed CBS platform [11,13,16,21]. However, a collective overview of the various available CBS in pharmacy practice education is lacking. This systematic narrative review aims to address this gap, describing the different computer simulators for pharmacy practice education, and highlighting some of their design characteristics.

## 2. Methods

Given the wide variety of trial designs and publication methods employed in the relevant literature (i.e., variation in terms of the study timeline, reported outcomes, focus, the CBS used, data gathering methods, and targeted participant group), a systematic narrative review method was employed [22,23,24] and the Preferred Reporting Items for Systematic Reviews and Meta-Analyses (PRISMA) guidelines were followed for reporting the findings [25]. 

### 2.1. Data Sources and Search Strategy

A systematic search was conducted using five major literature databases: Ovid Medline, CINAHL, ERIC, Education Source, and Ovid Embase, from January 2000 to August 2021. The bibliographies of the included studies were searched manually for the identification of any additional resources. The authors also explored the websites of the commercially available simulators which were identified *(MyDispense, DecisionSim, GIMMICS, Pharmacy Simulator, SimPharm, Virtual Interactive Case System (VICs), EHR GO).* In addition, the authors searched two Journals specialising in pharmacy education (American Journal of Pharmaceutical Education “AJPE” and Currents in Pharmacy Teaching and Learning “CPT&L”) as an extra measure to ensure the search was comprehensive. The search terms used in this study were “pharmacy education”; “teaching method”; “computer-based simulation”; “online simulation”; “virtual simulation”; and “virtual patient”.

### 2.2. Screening and Selection Criteria

Inclusion criteria were studies that: (a) discussed CBS in pharmacy practice education and (b) were published between January 2000 and August 2021. The exclusion criteria were publications that were either: (a) in languages other than English or (b) focused primarily on other types of simulations (e.g., role play, mannequins). 

The title, abstract and full text of each potentially relevant article was independently screened by two authors (AG and MS) for eligibility for inclusion, using “Covidence” [26]. Any discrepancies were resolved by a third author (IB/GP), and decisions were made by consensus.

### 2.3. Data Extraction and Synthesis

A full review of the included papers was completed by two reviewers (AG and MS) independently. Thematic analysis was carried out to extract the main themes relating to CBS design features [27]. The themes were categorised and discussed in the narrative synthesis of the literature.

All the extracted data were grouped in a tabular format that highlights: information about the study (authors, year, title); country; data collection methods/evaluation methods; participants; identified simulation tool; and the CBS’s described features (Appendix A). 

## 3. Results

The primary search using five databases identified a total of 1801 studies as being potentially relevant to this review. Of those, 43 were found to be eligible for full-text analysis. After full-text analysis, 5 more studies were excluded as they failed to meet the pre-defined inclusion criteria (not in English or focusing on other types of simulation), but another 11 potential studies were identified from the citation analysis. Ultimately, 49 studies were included (Figure 1). 

### 3.1. General Characteristics of the Included Papers

The majority were descriptive studies (83.7%; Appendix A). These studies generally measured self-efficacy, satisfaction, knowledge, application of concepts and knowledge retention. On the other hand, the qualitative studies focused on students’ attitudes or perceptions of their experience with CBS. The studies had a broad global representation: the United States of America (n = 23), Australia (n = 8), the UK (n = 5), New Zealand (n = 4), Canada (n = 2), Brazil (n = 2), Sweden (n = 1), Portugal (n = 1), Netherlands (n = 1) and two studies that examined participants from different countries (one in Australia and Malaysia, and the other for multiple European country graduates).

Many studies used online post-course/post-assignment surveys as the preferred data-gathering method; however, a few used pre- and post-surveys [4,16,28,29,30]. Other methods of data collection included focus groups [31,32,33,34,35], Subjective Objective Assessment Plans (SOAPs) [36,37], and observations [32,34,38,39,40]. Participants in the included studies were mostly pharmacy undergraduates, but some were postgraduate students or pharmacists.

Some studies focused on dispensing and communication skills development [12,31,41,42,43], while others focused on the assessment of the knowledge, metacognition support, and promoting active learning, student confidence and independent studying [4,11,13,15,16,28,33,35,36,37,40,44,45,46,47,48,49,50,51,52,53,54]. Some literature focused more on presenting the CBS platform and students’ perceptions of it [34,38,39,55,56,57,58,59,60,61,62,63,64,65]. A few studies assessed the use of CBS platforms against other training methods [16,32,40,51,61,63,66,67]. Others were primarily focused on case studies using electronic health records (EHR), such as ‘EHR GO!’ and ‘Virtual EHR’ [68,69,70]. 

### 3.2. Summary of Included Computer-Based simulators

The studies included in the literature search reported the use of 29 different simulators with various features, applications, and evaluation methods, as seen in further detail in supplementary data tables (Appendix A). Extracting information of interest for this paper regarding these simulators required some consolidation and further investigation. For example, two of the described simulators, *vpSim* and *DecisionSim*, were used interchangeably in the literature with little explanation about how they function, i.e., how users interact and how the virtual patient responds. Further exploration found they use the same multimedia-based software. One study provided the name of the CBS program used, but no information could be found regarding how users interacted with *PharmaCAL* [15]. Some of the described simulators were found to be outdated and no longer in existence, either because they were absorbed into a newer version (*Pharmville* database and resources included within *MyDispense*, *web-SP* into *Virtual Case Editor*—VCE [71]), or had been sold and discontinued, as in the case of *TheraSim* [72,73]. It was noted that some systems were not standalone software; rather, they were built upon existing generic educational platforms such as *Moodle* (https://moodle.com/, accessed on 23 march 2021) [50] or built within a commercially available platform such as *Second Life* (https://secondlife.com/. accessed on 23 march 2021) [40]. Moreover, one study used an EHR simulation *Virtual EHR/DocuCare,* primarily designed for nursing education as a technology demonstrator and proof of concept [74]. However, the simulator itself did not offer a specific training program for pharmacy students.

As a result, further investigation was carried out by visiting the CBS developers’ websites and searching for recent data regarding these simulators, as some studies dated back as far as 2001 [63] with no continuation of reported progress. It was found that eight simulators (out of the identified 29 simulators from the literature) are still available and actively offer their services for pharmacy education: *MyDispense* [75], *DecisionSim/VpSim* [76], *Pharmacy Simulator* [77], *Gimmics/PharmG* [78], *EHR GO* [79], *Keele virtual patient* (Keele University) [80], *Virtual Interactive Case System* (VICs) [81] and *SimPharm* [82].

Three simulators (*EHR GO*, *Virtual Interactive Case System* (VICs) and *DecisionSim/VpSim*) were originally designed for use in other disciplines but then adapted to use in the pharmacy setting. *Virtual Interactive Case System* (VICs) seems to be abandonware (for which no official continuous support or newer updates are available); however, it is still accessible at the time of writing.

Eight main criteria were used to examine the identified eight simulators (feedback type, grading, user play mode, cost, operational requirement, community/hospital setting, scenario sharing option, and interaction elements), as seen in Table 1. Five of those eight criteria were guided by the previous work of Choi and Milheim [83,84]. Those criteria were: (1) user play mode (single vs. multiple users)—whether a simulation is to be played by the individual learner, or in collaborative groups; (2) grading—whether the simulator can autonomously grade the student’s performance; (3) environment setting—the physical entities, topics, events, and environment that are being simulated (to adapt this concept more to the focus of this paper (pharmacy practice education), we used “community/hospital setting”); (4) interaction elements—how the student interacts with and receives sensory feedback from the simulation; and (5) feedback type—the way the simulator provides instructional assistance to users/students.

The three additional criteria were added by the authors, including (1) *scenario sharing option*—whether the simulator allows users to create and/or share scenarios outside of their organisation; (2) *cost*—which describes any fees associated with accessing the simulator; and (3) *operational requirement*—which describes the equipment and/or services required to access the simulator.

#### 3.2.1. Feedback

Students require feedback from which they can learn. We observed two main feedback mechanisms used in the reviewed papers: immediate or delayed. Immediate feedback is when the feedback that is given to the student in response to their action is provided contextually and immediately after finishing the exercise, without requiring a human assessor. Delayed feedback requires a facilitator to give feedback on the student’s performance. How timely this feedback can be influenced by several external factors (availability, time, and workload). Six out of the eight available simulators offered immediate feedback. 

#### 3.2.2. Grading

Grading and feedback are typically associated with each other. However, grading assigns a value to the student’s performance, while feedback provides the students with information about their choices and, where necessary, direction for improvement. The authors identified three approaches under this category: (a) automated grading (which typically comes with immediate feedback when this feature is supported), (b) needs facilitator marking or (c) does not support a grading feature. Five out of the eight active simulators offered automated grading, two simulators (*Pharmacy Game* and *EHR Go*) needed a facilitator to mark the exercise, and one simulator (*Keele virtual patient* (Keele University)) did not offer to grade [87].

#### 3.2.3. User Play Mode

CBS generally offer three user experience options: *single-user*, *multi-users*, or *both*. The *single-user* experience is where students can be assigned and play the whole scenario individually, at their own pace and time, without needing assistance from colleagues or facilitators. However, a *multi-user* experience is when a team of students with/without direct supervision from facilitators/educators are assigned to complete the same scenario, where each one has a specific role to play. 

Six of the eight active simulators had a single-user design focus. One simulator *EHR Go* offered both single option (one user independently completes the scenario) and multi-user customisation (group of users with/without a facilitator/educator) depending on the scenario design and aims, while another simulator (*Pharmacy Game*) focused solely on a multi-user design (user-collaboration intensive) where all activities require a group of users and facilitators to be in contact for the scenario to be completed.

#### 3.2.4. Cost

All eight CBS platforms are available for users, at varying costs. For example, *MyDispense* and *Virtual Interactive Case System* (VICs) are offered free of charge. In the case of *MyDispense*, the potential user needs to contact and apply through the *MyDispense* team for further set-up, while a limited number of scenarios are available for free with *Virtual Interactive Case System* (VICs). Both platforms, however, are limited in their capacity to train in different pharmacy settings (at the time of writing, *MyDispense* offers only the community setting and *Virtual Interactive Case System* (VICs) only the hospital setting). Other platforms have different setting pricing arrangements, e.g., *Pharmacy Simulator* has two options for pricing: individual lifetime access, or an institutional subscription plan on a per-user basis. *EHR Go* is offered free of charge for the educational institute; however, either the students purchase a subscription based on the amount of time they will need access, or the institutions choose to buy a subscription plan for their students.

#### 3.2.5. Operational Requirement

Seven of the eight simulators chose a relatively lightweight web-based design, with one of them (*Pharmacy Simulator*) also offering a downloadable standalone application for different devices (PC/Mac/Android/iOS), which provides a slightly enhanced user experience. On the other hand, one CBS (*Pharmacy Game*) relied on several different Microsoft applications to represent and play their simulated scenarios. All of the available simulators appear to operate well on typical business-grade computer hardware, with no unusual hardware requirements, although *Pharmacy Simulator,* as a fully 3D experience, does benefit from improved performance on higher specification devices. 

#### 3.2.6. Community/Hospital Setting

Some of the simulators (*Pharmacy Game*, *SimPharm*, and *MyDispense*) were focused on one practice setting, such as a community pharmacy or hospital, and showed a design focus that may not be readily expanded to allow for scenarios in other pharmacy practice settings or include interdisciplinary training. *Pharmacy Simulator* includes both community and hospital environments and scenarios, and also has a community doctor’s clinic environment available, although this was not being utilised at the time of writing. *DecisionSim/vpSim* and *SimPharm* are mainly used in a hospital setting as they focus more on Pharmacology training.

#### 3.2.7. Scenario Sharing Option 

Seven out of the eight simulators supported scenario-sharing features to different degrees. Only, the *Virtual Interactive Case System* (VICs) was unclear if it could share cases between users or across disciplines. The scenario bank feature is particularly emphasised by *Pharmacy Simulator*, *MyDispense,* and *SimPharm* thanks to their wider user network compared to other simulators and a lesser degree in *Keele virtual patient*, and *EHR Go.* Data regarding using such a feature in either Pharmacy Game or DecisionSim were unavailable. 

Also, it was noted that some simulators, such as *Pharmacy Simulator* and *MyDispense* support user-developed scenarios to be shared without any restrictions, while others require the developers’ input in building scenarios, as in the *Keele virtual patient*, or offer developer-controlled prebuilt cases repository, as in *SimPharm*.

#### 3.2.8. Interaction Elements

As seen in Table 1, five simulators offered a web-based forms design, that also included static images, text, patient notes and sometimes recorded video or audio (*DecisionSim/vpSim, MyDispense, SimPharm, Virtual Interactive Case System*” (VICs) and *EHR GO*). Only two simulators (*Pharmacy Simulator* and *Keele virtual patient* (Keele University)) used an interactive animated avatar output design (a three-dimensional model, which is capable of approximating the facial expressions and body language of an actual human in a given situation). With *Pharmacy Simulator*, both the characters in the scenario and the player are free to move around the virtual environment.

*Pharmacy Game* heavily relies on other team members ((human-to-human) interaction) using a relatively complex procedure of sharing and grading different steps through several Microsoft apps.

#### 3.2.9. Global Reach

It is also important to note that some simulators emphasise their ability to reach a wider network globally, as with *MyDispense, Pharmacy Simulator, Pharmacy Game* and *SimPharm*. This global reach is believed to boost the confidence in their product and show a greater ability in fulfilling different clients’ requirements and the ability to continue supporting updates and software fixes for their CBS.

*MyDispense* offers a much wider range of users across the world and is actively updating its content with the latest update “Version 7” offered at the “*MyDispense* global Symposium in July 2022 in Italy”.

## 4. Discussion

The scope of this systematic narrative review included a summarised overview of available CBS in pharmacy practice education, guided by a literature search and subsequent examination of relevant product websites. The study of CBS in pharmacy practice education has lagged behind some other health professions [22,59,88]. Although this review identified 29 simulators in pharmacy practice education, it is apparent that only eight are still active and in use today, and fewer still were readily available to integrate into the curriculum on normal commercial terms. 

The data collected from the studies and developers’ websites in this review showed some similarities among many of the CBS. Most advertised features such as being web-based, providing immediate feedback with automated grading, and being available for individual learners, with shareable scenarios. However, there were key differences to note. 

### 4.1. Operational Requirement 

Most of the available CBS in Table 1 used a web-based interface, requiring only a relatively basic internet-connected device. However, from trialling some of the complex animated avatar-based CBS (such as *Pharmacy Simulator* and *Keele virtual patient*), some slowness can be experienced depending on the internet connection or the specifications of the hardware being used to access the simulator. *Pharmacy Simulator* attempts to resolve this by also offering an optional standalone application version, which is more performant than the web-based experience. 

*Pharmacy Game* was the obvious outlier in this category, using a hybrid online delivery built around group interaction, with the need for extensive coordination between a team of users (facilitators and students). To facilitate the activity, they relied heavily on several separate Microsoft applications, such as Microsoft Teams, Microsoft Forms, Microsoft Excel and Microsoft Power Apps [85]. This may adversely affect staff time and effort required to deliver the exercise and introduces a dependency on these applications/services being available to the users—although presumably similar results could be achieved by selecting an appropriate set of competitor products. 

### 4.2. Immediate Feedback 

While delayed feedback that involves a facilitator may initially seem to provide a more personalised and detailed evaluation to students, it lacks standardisation and impacts educators’ workload and time, as they must individually assess each student’s work, year on year. 

The literature showed that students appreciated the immediate guidance and feedback provided, especially for repetitive training or preparing for exams, with the ability to undertake more activities and view results almost immediately. This promoted active learning and independent studying [4,13,15,16,28,35,36,52,53,54].

It has been demonstrated in other disciplines that immediate feedback enhances performance on examinations and promotes the retention of knowledge [89,90]. This also aligns well with the concept of a post-simulation debrief, which is emphasised in literature discussing simulation-based learning [91].

### 4.3. Automated Grading 

Automated grading is a popular feature among the eight available CBS platforms. Similar to the immediate feedback, reasons such as educators’ workload, time, and standardisation may be a strong motivation for the prevalence of automated grading in CBS design. However, it is also important to note that grading may not be an important feature, depending on one’s aim and needs. For example, some educators may focus primarily on ensuring their students are exposed to a greater range of scenarios for practice purposes, and not see any need to also assess their performance with grades, as this is already done elsewhere in the curriculum. This aligns with the design decision of the *Keele* university virtual patient team, which did not offer a grading feature (Richardson et al., 2019). 

### 4.4. Scenario Sharing Option 

User-generated and shared scenarios appear to have emerged as a necessary solution to the fact that developing scenarios can be complex and time-consuming. By providing this feature which allows educators to create and widely share scenarios independently (without support or intervention from software developers), it becomes possible to achieve a critical mass of scenarios that cover a broad range of topics, via a crowd-sourcing approach. If this is achieved, it becomes easier for any given educator to integrate the simulation into their curriculum, as they can borrow scenarios developed by various other educators, and will not be excessively burdened by having to develop them all independently [16,92,93], which improves teaching efficiency compared to traditional approaches involving standardised patients (Tai et al., 2020). The literature shows that such a collaborative feature positively impacts cost, time, and resources without compromising teaching quality [29,41].

### 4.5. User Play Mode

Six of the eight active simulators use a single-user design focus. The authors think that single or multi-user design preference should depend on the specific course’s aim and needs. Some courses may need either (a) individualised training with formal rules and structure, that suits each student’s own pace and time (hence preferring single-user design) or (b) more focus on social rules, communication and teamwork skills that need coordination and group interaction (hence preferring multi-user design).

This is in line with earlier work in educational simulation and gaming literature, where it was found that multiplayer games are process intensive and use social rules, but depending on the course aim, they can also be applied for open-ended, individual, and/or social learning [94]. 

### 4.6. Cost

While all eight platforms are available for use by other institutions to some degree, the cost of purchase and method of estimating the cost varies widely. Organisations must weigh up budget considerations against their operational need. It is also important to consider the overall budget consequences and not just the direct cost of the product. Will the purchase of a simulation tool reduce teaching/marking hours? What are the costs associated with generating or tailoring scenarios to suit your curriculum? Are there administrative overheads to consider in terms of technical support staff or any special hardware requirements? 

It is also worth noting that many of the products identified in this review did not report an easy mechanism for acquiring and integrating their product into pharmacy curricula, and some were not even responsive to email enquiries (Table 1). Only MyDispense, Pharmacy Simulator, SimPharm, and EHR GO had clearly defined pathways for licensing and integrating the product. 

### 4.7. Community/Hospital Setting

Various simulators focus on replicating one or more practice settings. This may be an important feature depending on the needs and aims of the individual organisation. If an organisation has needs across multiple disciplines or needs for a different training setting, they are likely to see efficiencies in purchasing one product that can service their needs, as products are typically licensed either per institution or per user, rather than per topic.

Some simulators, as seen in Table 1, show a specific speciality in working in either community or hospital settings or for a specific training like EHR training. However, this can change with the continuous specific customisation of CBS developers to meet their client needs. This is especially evident where for example, *Pharmacy Simulator* has a community doctor’s clinic environment available, but not utilised at the time of writing. It can also be evident where some simulators *MyDispense* and *Pharmacy Simulator* can offer EHR training if needed by their users.

### 4.8. Interaction Elements

Some simulators (such as *Pharmacy Simulator* and *Keele virtual patient*) offer an interactive animated avatar that brings life to the interaction between the user and the platform where facial expressions and tone (negative and positive effects) can guide the student into a more realistic experience. 

Further detailing options are offered in the *Pharmacy Simulator*, such as expressive patient responses according to mood and other parameters in multiple settings (including physiological modelling, vital signs, pain, etc.) [77]. This is an added complexity that other platforms lack. However, advanced customisation and realism come with a potential trade-off, as designing a scenario in *Pharmacy Simulator* may take more time and effort when the scenarios are more detailed and complex.

Another problem with highly realistic simulators is the students’ expectations. Some studies found that despite the simulated scenarios being closer to real-life than other contemporary learning tools, the appearance of the virtual avatar and its interaction experience with the user was still described by some participants as unrealistic [16,48,54,62,65]. This may relate to the concept of the “uncanny valley”—where paradoxically, as a representation of a human becomes closer to reality, the human observer can become more focused on the flaws [95].

### 4.9. Implications for Practice

Although a number of products have been developed and trialled for pharmacy practice education, relatively few CBS are active. Of the eight we identified as being available, only *MyDispense, Pharmacy Simulator, SimPharm*, and *EHR GO* demonstrated clearly defined pathways for licensing and integration.

Of these four products, each had some apparent strengths and weaknesses. MyDispense appears to be perhaps the most widely used internationally, and appeared to have the easiest scenario editing capabilities, although it is focused on relatively less complex community pharmacy patients, with fewer options for interaction. Pharmacy Simulator is a more interactive and graphically stimulating experience, with relatively more complex scenarios across both community and hospital pharmacy environments, but it has higher system requirements. Although the scenario editor is powerful, it is also more complex to learn how to use. SimPharm appears to have strengths in the area of clinical pharmacology, with special features focusing on pharmacokinetics. While EHR Go, with it’s collaborative multi-user EHR approach is perhaps the better choice if you want to familiarise students with the complexities of real patient health records in a collaborative environment, although we note that there are other competitor’s products for this particular purpose which did not make inclusion in this paper.

Despite the availability of these few products, it is apparent that global uptake of these types of learning tools is still relatively low. MyDispense reports around 200 partner institutions globally, while Pharmacy Simulator is used by only around 20 organisations. Although these numbers do not appear insignificant, the International Pharmaceutical Federation (FIP) list around 1830 Schools of Pharmacy. Furthermore, of the schools that do utilise these tools, it is not known how extensively these tools are integrated into their curriculum. 

The authors’ key recommendation to educators, in line with previous research recommendations [42,43], is that educators are encouraged to consider their own institutional, professional and curriculum needs, and then decide which platform best aligns with their teaching goals. Further studies are warranted to systematically explore the potential barriers that may impact the implementation of CBS, as identifying these issues is the first step to resolving them, so that future pharmacy educators may successfully implement a CBS.

### 4.10. Limitations

Although best efforts were made, extracting information about some of the CBS features or functions was difficult. Few of the described simulations have available free trial options, which limits the opportunity to experience the CBS first-hand and extract needed data when it is not available in published works. Conversely, since the creator of Pharmacy Simulator was a part of the research team, we had richer access to information about this product, which may have given it greater emphasis.

Also, when it comes to EHR training, the authors are aware that some pharmacy schools may have tested EHR training simulators that are not explicitly designed for pharmacy education, as a proof of concept or for a single module design; this was noted when, for example, the School of Pharmacy at the University of Pittsburgh incorporated a virtual EHR platform specific for nursing education (*DocuCare*, Lippincott Williams & Wilkins) [74]. Also, it is noted that other US pharmacy schools may have used other platforms (that are not pharmacy education specific) such as *NeehrPerfect*, *Cerner Academic Education Solutions*, *Epic*, *NiaRx, Elsevier Evolve*, *QS1*, and *a proprietary program* [96]. However, these simulators have not been thoroughly studied in the pharmacy education literature and do not specifically offer pharmacy training.

Moreover, realising the continual updating or adding of features, the authors acknowledge that this paper may not cover all of the simulators’ current potential, such as the new version 7 of *MyDispense* or the *Pharmacy Simulator* community doctor’s features, or its experimentation with other healthcare disciplines.

We also acknowledge that there may be simulators that we missed mentioning in this paper since they are newly introduced to the market, and some may not be reported in the literature or on developers’ websites yet, such as *KARE* and *KARE Diabetes Assessment* developed by the same team who developed the *Keele virtual patient* (Keele University).

## 5. Conclusions

We have provided an overview of available computer simulators for pharmacy practice education, guided by a comprehensive literature search. This review provides educators with a deeper insight and understanding of the available state-of-the-art computer simulators. When considering the adoption of computer simulators in pharmacy practice education, educators are encouraged to develop a clear understanding of their specific institutional needs and weigh these up against the features and functionality described herein.

## Figures and Tables

**Figure 1 pharmacy-11-00008-f001:**
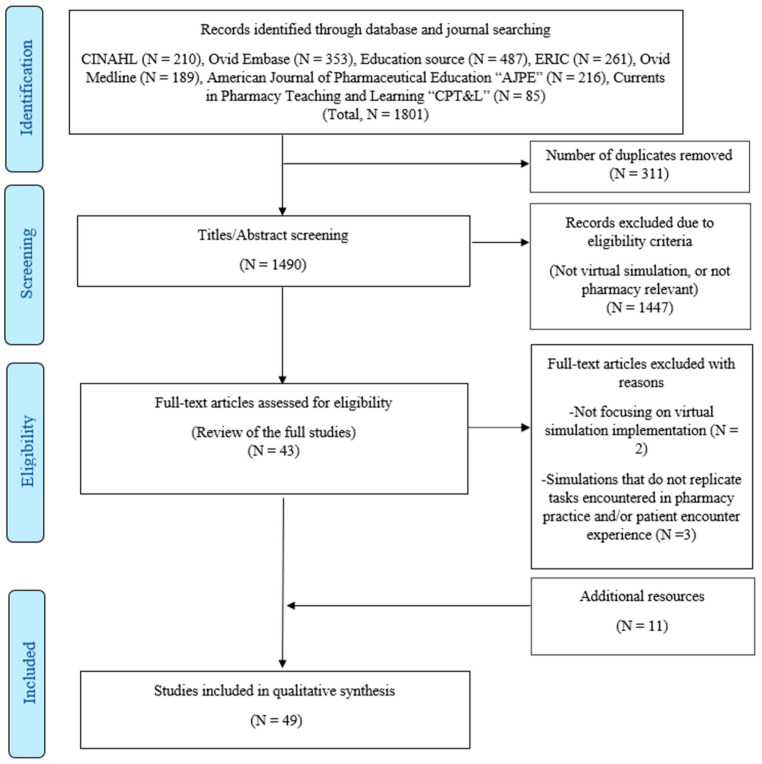
Flow diagram of the study selection process.

**Table 1 pharmacy-11-00008-t001:** Characteristics of the eight available computer-based simulators.

Simulator Name and Country of Origin	Feedback Type	Grading	User Play Mode	Cost	Operational Requirement	Community/Hospital Setting	Scenario Sharing Option	Interaction Elements
DecisionSim/ vpSim [76] USA	Immediate	Yes, Automated	Single	Cost not advertised ^a^	Web-based	Hospital ^b^ (Pharmacology focus)	Yes	Video, audio, and images (Web-based)
MyDispense [75] Australia	Immediate	Yes, Automated	Single	Free of charge	Web-based	Community	Yes	Static images, text, and patient notes (Web-based)
Pharmacy Simulator [77] Australia	Immediate	Yes, Automated	Single	Cost is publicly available ^c^	Web-based or standalone application (mobile/desktop)	Community and Hospital	Yes	Interactive animated avatar s with voiced dialog and dynamic emotions, patient notes, health record, dispensing, vital signs monitoring, administration of medicines, hand washing, navigating the environment (3D game-like virtual environment)
Keele virtual patient (Keele University) [80] UK	Immediate	No grading	Single	Cost not advertised ^d^	Web-based	Community/primary care pharmacist ^e^	Yes	Interactive animated avatar with voiced dialog and dynamic emotions, patient notes ^f^ (Static 3D virtual environment)
SimPharm [82] New Zealand	Immediate	Yes, Automated	Single	Cost not advertised ^g^	Web-based	Hospital (Pharmacology focus)	Yes	Static images, audio, text, and patient notes (Web-based)
Virtual Interactive Case System (VICs) [81] Canada	Immediate	Yes, Automated	Single	Free ^h^	Web-based	Hospital ^i^	Not-clear	Static image and text, with pre-defined choices (Web-based)
Pharmacy Game Previously known as “GIMMICS/ PharmG” [78,85,86] Netherlands/Australia	Delayed	Marked by faculty	Multi	Cost not advertised ^j^	Relies on other applications such as: Microsoft Teams Microsoft Forms Microsoft Excel Microsoft Power App	Community	Yes	Utilising Microsoft applications that allow sharing documents, audio, and video calls (Human-to-human interaction)
EHR GO [79] EHR GO® is a product of archetype innovations, LLC	Delayed	Marked by faculty	Single and Multi	Cost not advertised ^k^	Web-based	Community and Hospital ^l^	Yes	Static images, text, and patient notes (Web-based)

^a^ Further inquiry to the developers and marketing team was not successful (No reply received). ^b^ DecisionSim/vpSim has also been used in other disciplines, such as medical education, or community health worker training. ^c^ Further inquiry from the marketing team showed that prices typically start at $20 USD per student per year (depending on the number of users and the license duration). ^d^ Further inquiry from the marketing team showed that they do not use a per-user system for licensing. Also, they stated that most of their clients request a new virtual patient from scratch, which will cost around £15,000 to develop and includes a 12-month license. After that initial license period ends, an annual setup which is around £2000 per year with a discount if a client has multiple cases. ^e^ In Keele virtual patient, the user can be either a GP or a primary care pharmacist. ^f^ A non-interactive scenario has also been developed where users can answer in a free-text style and the feedback can be provided by displaying the ‘model’ answer for each of the questions, also, an explanation of the correct answer. ^g^ Further inquiry from the marketing team showed that prices typically start at $50 USD per student for an average batch of 100 students. ^h^ Pharmacy scenarios seem free and publicly available yet in limited numbers, specific to certain focuses, and not updated. In general, VICs is available at a cost for interested institutes where they can acquire a site license for the VIC editor and player for around $5000 CAD per site. ^i^ VICs can be used for training in other healthcare professions such as nursing, family medicine and anesthesia. ^j^ Further inquiry to the marketing team was not successful (No reply received). ^k^ EHR GO is free for faculty and the institution. Students purchase a subscription based on the amount of time they need access to the platform. Some institutions choose to buy Go for their students and offer a bulk discount for large purchases. ^l^ EHR GO can be used for other healthcare training programs such as interprofessional, nursing, dietetic.

## Data Availability

Not applicable.

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
