# Peer review of "Computer-Based Simulators in Pharmacy Practice Education: A Systematic Narrative Review"

_pharmacy, 2023, doi:10.3390/pharmacy11010008_

Round 1

Reviewer 1 Report

Thank you for your contribution.  Overall this is a well written manuscript that provides readers with an interesting overview of a fast-changing techology.  The authors describe a robust methodology that instills confidence in appropriate use of systematic review processes to ensure adequate coverage of the literature.  I note there are several instances where references appear to be missing or where 'error' messages appear in the text version I review.  Further, the tables require reformatting...the version that appeared on my computer screen had very peculiar spacing problems that made deciphering the content challenging.  I am assuming these formatting issues can be readily addressed so will not focus further attention on them.

My main recommendation is to provide some more comprehensive and informed analysis of findings.  The authors have done a strong job of the systematic analysis portion of the paper, but I find the discussion and conclusions a bit of a letdown.  The authors rightly conclude this is a fast-changing area so even this manuscript will be out of date by the time it comes to press.  Still, some perspectives on application to undergraduate, postgraduate, and CPD education would be helpful, and a more sophisticated discussion of gaps across all these platforms would be interesting.  Comparisons to other professions, or signposting of future trends/directions in simulation would also be useful to include.  I'm not suggesting all of these be added to the mansucript, but some kind of summation by the authors would help round out the paper and simply make it more useful for readers.  As a systematic review it works effectively and is well written; for it to be useful for readers especially educators, more context and summation would be helpful.

Author Response

Reviewer #1:

Comment: Thank you for your contribution. Overall, this is a well-written manuscript that provides readers with an interesting overview of fast-changing technology.  The authors describe a robust methodology that instils confidence in the appropriate use of systematic review processes to ensure adequate coverage of the literature. I note there are several instances where references appear to be missing or where 'error' messages appear in the text version I review. Further, the tables require reformatting...the version that appeared on my computer screen had very peculiar spacing problems that made deciphering the content challenging.  I am assuming these formatting issues can be readily addressed so will not focus further attention on them.

Response: Thank you for the suggestions.

We apologise, it appears the formatting was corrupted when we converted the original manuscript to use the MDPI style. We have repaired the damaged tables and reference list in the revised manuscript.

Comment: My main recommendation is to provide a more comprehensive and informed analysis of the findings.  The authors have done a strong job of the systematic analysis portion of the paper, but I find the discussion and conclusions a bit of a letdown. The authors rightly conclude this is a fast-changing area so even this manuscript will be out of date by the time it comes to press. Still, some perspectives on application to undergraduate, postgraduate, and CPD education would be helpful, and a more sophisticated discussion of gaps across all these platforms would be interesting. Comparisons to other professions or signposting of future trends/directions in a simulation would also be useful to include.  I'm not suggesting all of these be added to the manuscript, but some kind of summation by the authors would help round out the paper and simply make it more useful for readers. As a systematic review, it works effectively and is well written; for it to be useful for readers, especially educators, more context and summation would be helpful.

Response: Yes, thank you. We agree. In the revised manuscript, the implications for practice in section 4.9 have now been amended and strengthened according to the valued feedback received. Perspectives on the implication for practice have been added, along with a more refined summation and a recommendation on how educators should approach the implementation of CBS.

Reviewer 2 Report

Useful review of the state-of-the-art CBS for pharmacy training. There are two minor points:

1. Criteria for decision to select (only) 2 journals, specializing in pharmacy education are missing. Why not also Pharmacy?

2. Not all references are written in the text and the Table need to be graphically improved 

Author Response

Reviewer #2: 
Comment: A useful review of the state-of-the-art CBS for pharmacy training. There are two minor points:

1.    Criteria for the decision to select (only) 2 journals, specializing in pharmacy education are missing. Why not also Pharmacy?

Response: We want to clarify that our systematic review search employed five major databases (Ovid Medline, CINAHL, ERIC, Education Source, and Ovid EMBASE), and subsequently, we also manually searched two key pharmacy education-focused journals (American Journal of Pharmaceutical Education (AJPE), and Currents in Pharmacy Teaching and Learning (CPT&L)), This was only done as a precaution, to capture any further potential papers. However, we can see how reporting this step has caused some confusion; therefore, we decided to omit it from the main manuscript, especially since there were no significant studies included in that step. 

2. Not all references are written in the text and the table needs to be graphically improved

Response: Apologies. We believe the formatting was corrupted when we converted the original manuscript to MDPI style. We have now repaired the tables and references in the revised manuscript.